# Sex-Specific Cut-Off Values for Low Skeletal Muscle Mass to Identify Patients at Risk for Treatment-Related Adverse Events in Head and Neck Cancer

**DOI:** 10.3390/jcm11164650

**Published:** 2022-08-09

**Authors:** Aniek T. Zwart, Wolf Pörtzgen, Irene van Rijn-Dekker, Grigory A. Sidorenkov, Rudi A. J. O. Dierckx, Roel J. H. M. Steenbakkers, Inge Wegner, Anouk van der Hoorn, Geertruida H. de Bock, Gyorgy B. Halmos

**Affiliations:** 1Department of Epidemiology, University Medical Centre Groningen, 9700 RB Groningen, The Netherlands; 2Department of Radiology, University Medical Centre Groningen, 9700 RB Groningen, The Netherlands; 3Department of Otolaryngology and Head and Neck Surgery, University Medical Centre Groningen, 9700 RB Groningen, The Netherlands; 4Department of Radiotherapy, University Medical Centre Groningen, 9700 RB Groningen, The Netherlands

**Keywords:** sarcopenia, head and neck neoplasms, adverse events, postoperative complication, toxicity, cut-off, skeletal muscle mass

## Abstract

A low skeletal muscle index (SMI), defined with cut-off values, is a promising predictor for adverse events (AEs) in head and neck squamous cell cancer (HNSCC) patients. The aim was to generate sex-specific SMI cut-off values based on AE to diagnose low SMI and to analyse the relationship between low SMI and AEs in HNSCC patients. In this present study, HNSCC patients were prospectively included in a large oncological data-biobank and SMI was retrospectively measured using baseline neck scans. In total, 193 patients were included and were stratified according to treatment modality: (chemo-)radiotherapy ((C)RT) (n = 135) and surgery (n = 61). AE endpoints were based on the occurrence of clinically relevant toxicities (Common Terminology Criteria for Adverse Events grade ≥ III) and postoperative complications (Clavien–Dindo Classification grade ≥ II). Sex-specific SMI cut-off values were generated with receiver operating characteristic curves, based on the AE endpoints. The relationship of the baseline characteristics and AEs was analysed with logistic regression analysis, with AEs as the endpoint. Multivariable logistic analysis showed that low SMI (OR 3.33, 95%CI 1.41–7.85) and tumour stage (OR 3.45, 95%CI 1.28–9.29) were significantly and independently associated to (C)RT toxicity. Low SMI was not related to postoperative complications. To conclude, sex-specific SMI cut-off values, were generated based on the occurrence of AEs. Low SMI and tumour stage were independently related to (C)RT toxicity in HNSCC patients.

## 1. Introduction

Worldwide, head and neck cancer incidence affects 650,000 people and leads to 330,000 deaths annually [1]. In general, head and neck cancer is a heterogeneous group with upper aero-digestive-tract malignancies, with most patients having squamous cell carcinoma of the head and neck (HNSCC) [2]. Surgery, radiotherapy and/or systemic treatment are the cornerstones of HNSCC treatment. As most patients with HNSCC are diagnosed with advanced stages of disease, intensive treatment is necessary, which is associated with a high prevalence of adverse events (AEs) [3,4]. AEs can lead to early treatment termination, resulting in higher mortality, and may impact patients’ quality of life [5]. Therefore, it is essential to predict which HNSCC patients will develop adverse events, to make a proper treatment selection and/or to optimise patients before treatment.

In the last decade, a low amount of skeletal muscle mass has been found to be a promising radiological biomarker for adverse events in HNSCC patients including (chemo)radiotherapy ((C)RT) toxicity, postoperative complications and mortality [6,7,8,9]. Decline of muscle is a biological process, but HNSCC patients are susceptible to muscle wasting due to (pre-existent) inadequate nutrition and cancer cachexia [10]. The latter is defined as a complex metabolic condition, characterised as an inflammatory process, with muscle loss, which is most prevalent in metastatic disease. Skeletal muscle mass is measured with skeletal muscle index (SMI) using abdominal imaging in oncology patients [11,12]. However, HNSCC patients generally do not have abdominal imaging, except in exceptional cases in the setting of a metastatic workup. Therefore, SMI has been recently measured using neck imaging [13,14]. This enables measuring SMI in an HNSCC population without abdominal imaging. To define which HNSCC patients have low SMI, SMI cut-off values have to be applied. Although no consensus has been reached about which SMI cut-off value to use, it is recommended to apply sex- and region-specific SMI cut-off values, as SMI is known to differ between sex, ethnicity and cancer types [15,16,17,18,19]. The current literature addressing SMI cut-off values based on the occurrence of toxicities or postoperative complications in HNSCC has some limitations, as these are not sex-specific [20,21] or were they generated in an Asian population [6,22,23,24,25]. Moreover, previously reported sex-specific SMI cut-off values in a Western cohort of HNSCC of van Rijn-Dekker et al. and Karavolia et al. were based on the lowest interquartile range [26,27]. This method is seen as inferior, as it may underestimate the effect of low SMI on the occurrence of adverse events.

A more population-fitted SMI cut-off value using sex-specific SMI cut-off values, based on the occurrence of toxicities and postoperative complications to diagnose low SMI, could help identify HNSCC patients at risk for toxicity and postoperative complications. Moreover, when low SMI is found to be a good predictor, this may lead to (nutritional or exercise) interventions to prevent muscle wasting and, therefore, reduce the occurrence of toxicity and postoperative complications in HNSCC patients. The latter is underscored by a previous interventional study, as it was possible to maintain skeletal muscle mass with preoperative exercise in patients with pancreatic cancer [28]; however, such interventional studies in HNSCC patients does not exist yet. Thus, the role of interventional studies in maintaining or improving skeletal muscle mass in HNSCC remains unclear, although pre-treatment interventions were found to be related to an improved quality of life [29] and were found to be feasible [30]. Therefore, the aim of the present study was to generate sex-specific SMI cut-off values based on toxicity and postoperative complications, to diagnose low SMI and to analyse the relationship between low SMI and AEs in HNSCC patients.

## 2. Materials and Methods

### 2.1. Study Design and Ethical Approval

This retrospective cohort study is registered at the University Medical Center Groningen (UMCG) Research Register (number 201900587). Data were extracted from the OncoLifeS data-biobank with permission of the scientific committee of OncoLifeS [31]. The OncoLifeS data-biobank is approved by the by the Medical Ethical Committee of the University Medical Centre of Groningen and is registered in the Dutch Trial Register, registration number: NL7839. The OncoLifeS data-biobank collects data on adult oncological patients, including patient, cancer and treatment characteristics. Patients were prospectively included into the data-biobank, after patients’ informed consent.

### 2.2. Patient Population

HNSCC patients with primary tumours of the oral cavity, larynx, oropharynx, nasopharynx and hypopharynx, regardless of tumour stage, with pre-treatment neck imaging and with treatment with curative intent, were included. Modalities of primary treatment were: radiotherapy with or without neoadjuvant chemotherapy and surgery. In case of post-operative (C)RT, surgery was selected as primary treatment modality, and only the postoperative complications were included. Exclusion criteria were patients with a history of previous HNSCC, multiple simultaneous (head and neck and non-head and neck) malignant tumours, early-stage tumours (including stage I–II HNSCC and carcinoma in situ) treated by transoral laser surgery (because of its minimally invasive nature), treatment outside the UMCG and an interval of more than 30 days between pre-treatment imaging and treatment. SMI measurements were found to be unreliable in the case of insufficient imaging and, therefore, were excluded from the analysis. Insufficient imaging was categorised as too much angulation of the neck, insufficient signal to noise ratio or imaging artefacts, tumour muscle infiltration and (partially) not imaging the relevant anatomy. Patients were divided into groups according to their primary treatment: (C)RT or surgery. Surgical procedures were further stratified into minor and major surgery. Transoral surgery with or without sentinel node procedure were considered minor. Major surgery was in the case of commando procedure, total laryngectomy (TLE), hemi- or partial maxillectomy and transoral surgery with any type of neck dissection or flap reconstruction. In general, type of (C)RT treatment depended on oncological stage and patients’ age or fitness. Treatment options according to this protocol were: conventional fractionated radiation therapy (5 fractions per week); accelerated radiation therapy (6 fractions per week); accelerated radiation therapy with weekly cetuximab (400 mg/m^2^ 1 week before radiation therapy, followed by weekly 250 mg/m^2^ during radiation therapy); and concomitant chemoradiation (3 courses of carboplatin on day 1, 300–350 mg/m^2^ and 5-fluorouracil on day 1–4, 600 mg/m^2^/24 h every 3 weeks or 3 courses of cisplatin (100 mg/m^2^) on day 1, 22 and 43) [32]. HNSCC patients in the here-presented study were treated according to (inter)national guidelines within the multidisciplinary head and neck tumour board [4].

### 2.3. Baseline Characteristics

Baseline characteristics were derived from the OncoLifeS data-biobank, and, if needed, supplemented with data from the patients’ electronic medical files. The following characteristics were retrieved: age (years), sex, tumour site and stage, human papillomavirus (HPV) status, treatment modality, Body Mass Index (BMI, kg/m^2^), severity of comorbidities, frailty status, history of smoking and alcohol consumption. The 7th edition of American Joint Committee on Cancer Manual was active at time of prospective inclusion of patients and was used for oncological staging [33]. Comorbidities were evaluated with the Adult Comorbidity Evaluation-27 (ACE-27). Frailty was assessed with Geriatrics 8 (G8) and Groningen Frailty Indicator (GFI). Patients were defined as frail when G8 ≤ 14 or GFI ≥ 4 [34,35]. Smoking was expressed in categories “current/history of smoking” or “never smoked”. Alcohol was measured in terms of “abuse” or “no abuse”. Abuse was defined as >3 units per day for men and >2 units a day for women [36].

### 2.4. Skeletal Muscle Mass Analysis

Both CT and MRI neck scans were used to quantify skeletal muscle mass according to validated methods, as described earlier [13,14]. Shortly, delineation of skeletal muscle was done at level of the third cervical vertebrae (C3) including the right and left sternocleidomastoid muscle and paravertebral muscles, resulting in the cross-sectional area (CSA, cm^2^) at C3 level. Using CSA at C3 level, CSA at L3 level was estimated according to the prediction model of Swartz et al., and, with that, the patient’s SMI was calculated (cm^2^/m^2^) [13]. Delineation was performed with Aquarius workstation iNtuition edition program (ver. 4.4.13.P6, Terarecon Inc., Foster City, CA, USA). The main observer (W.P.), a medical student, performed all skeletal muscle measurements and was first trained in a training set, and the observer’s performance was tested before doing measurements in this study cohort. This was done according to previous published methods [14,37]. 

In case of both available imaging modalities, CT imaging was preferred because delineation of skeletal muscle is semi-automatic for CT. For CT, to ensure a good differentiation between muscle and other structures, scans were preferably post-contrast (n = 44), reconstructed with a 1 mm slice thickness (n = 25), and a soft-tissue kernel (n = 18). For MRI, neck scans were generated by 1.5 or 3 Tesla MRI scanners, and multiple sequences were available. Preference was made to perform skeletal muscle mass analysis on T2 weighted images, as this sequence has the highest agreement with CT neck imaging and has excellent observer reliability [14]. 

### 2.5. Endpoints for Adverse Events

After inclusion, patients were systematically followed up and the occurrence of AEs were prospectively recorded into the OncoLifeS data-biobank. Clinically relevant (C)RT toxicity was classified according to the Common Terminology Criteria for Adverse Events (CTCAE), with grade ≥ III as clinically relevant [38]. Toxicity was scored 12 weeks after the start of treatment. Domains scored by the CTCAE were: xerostomia, taste, throat pain, oral pain, general pain, dysphagia, hoarseness and mucositis. All-cause (C)RT toxicity was defined as CTCAE toxicity grade ≥ III, irrespective of domain. Dysphagia toxicity with CTCAE grade ≥ III was also scored separately.

Postoperative complications were classified according to the Clavien–Dindo classification (CDC), with grade ≥ II as clinically relevant [39]. Postoperative complications were measured as all-cause postoperative complications and as wound infection, independently. When multiple postoperative complications or toxicities occurred, the highest-graded postoperative complication or toxicity was selected as AE endpoint.

### 2.6. Statistical Analysis

Kolmogorov–Smirnov normality test was used for continuous data. Baseline descriptive statistics were presented as mean with standard deviation (SD) or median with interquartile range (IQR) for normal and non-normal distributed variables, respectively, and frequencies with percentages for categorical variables. Additionally, differences between patients treated with primary (C)RT and surgery were evaluated by the Student’s *t*-test, Mann–Whitney U test, chi-squared test and Fisher’s exact test, depending on the normality and type of data. Inter-observer reliability was determined by the inter-class correlation. To determine sex-specific SMI cut-off values based on the adverse events endpoints, a Receiver Operator Characteristic (ROC) test was performed, and the area under the curve (AUC), specificity and sensitivity of the cut-off value were provided. An AUC of ≥0.7 was considered as diagnostic acceptable [40]. Based on the four AE endpoints, eight sex-specific SMI cut-off values were generated. The Youden index was used to identify the optimal SMI cut-off values. To analyse the relationship of the baseline characteristics on the occurrence of adverse events endpoints, univariable and multivariable binary logistic regression were performed. Odds Ratios (OR) and 95% Confidence Intervals (CI) were provided. Clinically relevant and significant high impact variables from the univariable logistic regression were selected as covariates for the multivariable logistic regression, and covariates were excluded from the multivariable model in a backwards manner. A two-sided *p*-value of <0.05 was considered significanttistical analysis was performed using IBM SPSS Statistics (IBM Corp. Released 2021. IBM SPSS Statistics for Windows, Version 28.0. Armonk, NY, USA: IBM Corp).

## 3. Results

### 3.1. Patients’ and Baseline Characteristics

Between June 2013 and November 2019, 557 head and neck patients treated with curative intent at the UMCG were enrolled in the OncoLifeS data-biobank [31]. Of these patients, 224 were initially found suitable for inclusion in this retrospective cohort study, based on in- and exclusion criteria (Figure 1). Thereafter, patients were excluded as

Some patients had to be excluded from the initial study cohort due to imaging or muscle delineation difficulties, including too much angulation of the neck (n = 16), insufficient signal to noise ratio or artefacts (n = 11), tumour muscle infiltration (n = 12), and relevant anatomy was not or partially imaged (n = 2). Moreover, we had to excluded an additional nine patients as there were no data available on the endpoints. In total, 196 patients were included in the present study, of which 49 patients were also included in previous studies [14,37]. Of patients, 61 and 135 patients were respectively treated with surgery and (C)RT. Of the surgical group, 29 patients had minor surgery (transoral procedure with or without sentinel node excision n = 29) and 32 patients had major surgery (commando procedure n = 15, transoral procedure with any kind of neck dissection or flap reconstruction n = 10, total laryngectomy with or without neck dissection n = 4 and hemi- or partial maxillectomy n = 3). Of the (C)RT group, 74 patients received radiotherapy alone, 54 patients received chemoradiation and 7 patients had accelerated radiotherapy with cetuximab.

Baseline characteristics of the total cohort and differences between the two treatment modalities are presented in Table 1. Most patients were male (65.3%), and the mean age of the cohort was 64 (±11) years. The most common anatomical site of the primary tumour was the oropharynx (35.2%), followed by the oral cavity (30.6%), larynx (28.6%), hypopharynx (5.1%) and nasopharynx (0.5%). The cohort had a relatively advanced stage of disease, as most common tumour stage was stage IV (57.6%). Differences in baseline characteristics between the two treatment modalities are presented in Table 1.

### 3.2. Occurrence of Adverse Events

Occurrence of the adverse endpoints are displayed in Table 2. Thirty-three (24.4%) patients experienced clinically relevant toxicity (CTCAE ≥ III), irrespective of toxicity domain. On all toxicity domains, no grade IV or V (death) toxicity occurred. Twenty-nine (21.5%) patients suffered dysphagia grade III or more. Prevalence of CTCAE grade ≥ III on other toxicity domains were 3.0% xerostomia, 3.0% hoarseness, 2.3% mucositis, 2.3% general pain, 0.8% oral pain and 0.8% throat pain. No grade ≥ III toxicity on the domain of taste occurred.

Clinically relevant postoperative complications (CDC ≥ II) occurred in 21 (34.4%) patients. The highest scored grade was grade III, no grade IV or V (death) complication occurred. Prevalence of post-operative wound infection with CDC ≥ II was 16.9%.

### 3.3. Observer Analysis, Skeletal Muscle Quantification and SMI Cut-Off Values

Performance of the main observer for skeletal-muscle quantification in the teaching file was determined to be excellent as ICCs were r = 0.991 (*p* < 0.001) and r = 0.975 (*p* < 0.001) for MRI (n = 10) and CT (n = 10), respectively. Delineation of skeletal muscle used for further analysis was more often performed on CT (78.2%) than MRI (21.8%). For the total cohort, male patients had significantly higher SMI than female patients (47.6 ± 5.7 cm^2^/m^2^ versus 35.3 ± 6.5 cm^2^/m^2^, *p* < 0.001). Sex-specific SMI cut-off values were generated to identify patients having low SMI based on the adverse events endpoints. In total, eight SMI cut-off values were generated and sensitivity, specificity and AUCs were provided per SMI cut-off value (Table 3). AUCs were all <0.7 (0.425–0.688). ROC curves showing the diagnostic accuracy for sex-specific SMI and the occurrence of all-cause toxicities and postoperative complications are shown in Figure 2.

### 3.4. Relationship between Low Skeletal Muscle Mass and Adverse Events

For total cohort, prevalence of low SMI was 34.7% (3.9% of males, 92.6% of females), 34.7% (3.9% of males, 92.6% of females), 33.2% (4.7% of males, 86.8% of females) and 29.6% (0% of males, 85.3% of females) for SMI cut-off values based on all-cause toxicity, dysphagia, all-cause postoperative complications and wound infection, respectively. Although AUCs were <0.7 for all generated SMI cut-off values, a logistic-regression analysis was performed to analyse the relationship between low SMI and the adverse events endpoints.

An univariate logistic regression was performed with the AE endpoints as dependent variable (Table 4 and Table 5). For the multivariable regression analyses for toxicity, tumour site, oncological stage and low SMI were selected from the univariate logistic-regression analysis, according to their OR and *p*-values, and age was added to the model as a clinically relevant variable. Other potential important clinically relevant variables such as G8, GFI and ACE-27 were not included to the multivariable model due to a relatively high rate of missing’s. Moreover, BMI and gender were also not selected as these variables we used to calculate SMI. Tumour site and age were excluded from the model when backward selection was applied. Stage IV disease (OR 3.45, 95% CI 1.28–9.29) and low SMI (OR 3.33, 95% CI 1.41–7.82) remained independently and significantly related to all-cause relevant (C)RT toxicities (Table 6). The separate univariable and multivariable regression analysis (Table 4 and Table 6, respectively) for grade III dysphagia had similar results compared to the analysis with all clinically relevant (C)RT toxicities, as stage IV disease (OR 3.47, 95% CI 1.21–9.92) and low SMI (OR 2.50, 95% CI 1.02–6.14) were also independently and significantly related to grade III dysphagia. The univariable logistic regression for all-cause postoperative complications (CDC ≥ II) did not identified variables significantly associated with the endpoint (Table 5). Tumour site was the only variable significantly associated with the occurrence of wound infections (CDC ≥ II). Compared to the oral cavity, tumours located in the hypopharynx/larynx had the highest risk (OR 10.29, 95% CI 1.45–72.81) of developing postoperative wound infection.

## 4. Discussion

This is the first study that established both sex and treatment modality stratified SMI cut-off values based on clinically relevant adverse outcomes to diagnose low SMI in HNSCC patients. Analysis showed that SMI was significantly higher in males than females, in both the (C)RT group and the surgery group, supporting the hypothesis that sex-specific SMI cut-off values are mandatory to diagnose low SMI. Sex-specific SMI cut-off values were generated with ROC curves for the different AE endpoints. AUCs were all <0.7, indicating insufficient discriminative power of low SMI on the endpoints. Nevertheless, low SMI and stage IV disease were independently and significantly related to all-cause toxicities and dysphagia. However, no other characteristics significantly related to postoperative complications could be identified. Low SMI, according to the established sex and modality specific SMI cut-off values, is, therefore, related to all-cause toxicities and dysphagia but not to postoperative complications in this present cohort with HNSCC.

Currently, no consensus has been reached to diagnose low SMI in HNSCC patients. Skeletal muscle mass can be measured using different modalities including dual-energy X-ray absorptiometry, bioelectric impedance analysis and CT and MRI imaging [15]. CT and MRI imaging are often used in oncology as these images are already part of the normal oncological work-up, and, therefore, no additional scans have to be made. In CT and MRI imaging, the gold standard is to measure SMI at the L3 level [15]. However, abdominal imaging is often lacking in HNSCC, and neck CT or MRI at the level C3 were found to be a valid alternative to measure SMI [13,14]. However, other possible alternatives exist to measure SMI in HNSCC patients, for instance measurements on the masticatory muscles [41]. In general, SMI cut-off values can be generated using different methods including cohort mean (using -2SD), cohort median (using lowest interquartile range) and optimal stratification based on clinical outcome (using ROC curves or log likelihood models). The ROC method is the most optimal method of defining a clinical implacable cut-off value, since this method searches for the point where the sensitivity and specificity is the highest [40].

SMI cut-off values are often applied from Prado et al. and Martin et al. in oncology patients [42,43]. These cut-offs are based on the log likelihood of survival, and both studies included patients with solid tumours of the respiratory or gastrointestinal tract. SMI cut-off values of Prado et al. are remarkably higher compared to this present study. The difference can be explained as the cohort of Prado et al. consisted of mostly men, sarcopenic obese patients and a different oncological population. Apart from sex-specific SMI cut-off values, Martin et al. also introduced BMI-specific SMI cut-off values. The SMI cut-off values of Martin et al. are quite similar, especially for underweight and normal weight patients, indicating that HNSCC patients are normal- to underweight compared to their study population. This difference emphasises that HNSCC patients are in need of their own SMI cut-off value to diagnose low SMI.

Nevertheless, already published SMI cut-off values generated in a HNSCC population have several limitations as most are not sex-specific [21,44], are based on the median SMI of the given cohort [26,27], or have a specific (Asian) ethnicity [6,22,23,24,25]. Only three studies concerning toxicity in HNSCC patients have a Western study population [21,26,27] and are, therefore, comparable with the SMI cut-off values as generated in this present study, as SMI will differ between ethnicities. Wendrich et al. created a non-sex-specific cut-off value of <43.2 cm^2^/m^2^, using the lowest log-likelihood value based on chemotherapy dose-limiting toxicity in a Dutch population, arguing no difference in chemotherapy dose-limiting toxicity between males and females [21]. This seems controversial as female patients have lower SMI, resulting in low SMI of almost all female patients. As of today, Karavolia et al. provided sex-specific SMI cut-off values (SMI <42.0 cm^2^/m^2^ and SMI <31.2 cm^2^/m^2^ for males and females, respectively) in the largest published HNSCC population undergoing (C)RT [27]. Van Rijn et al. used very similar SMI cut-off values using the same strategy to generate SMI cut-off values; however, the cohorts of these studies are overlapping. Nevertheless, the established sex-specific SMI cut-off values of Karavolia et al. and van Rijn et al. are based on the lowest quartile, and this method is inferior, as it may underestimate the effect of low SMI on clinical outcome.

Currently, low SMI is gaining more interest and is increasingly recognised for its predictive value for adverse events in oncology. However, the relationship of low SMI and the occurrence of toxicity seems to be less investigated in HNSCC patients, as only four studies could be identified [26,27,45,46]. The study of Lere-Chevaleyre et al. showed that pre-treatment SMI was the only predictive factor of all-cause toxicity (CTCAE ≥ III) in head and neck cancer patients receiving induction chemotherapy by docetaxel, cisplatin and 5-fluorouracil [45]. However, Thureau et al. could not link low SMI with toxicity in HNSCC patients treated with chemotherapy or chemoradiation [46] and is, therefore, not in line with the results of this present study. The association between low SMI and dysphagia is supported by Karavolia et al. and van Rijn et al. [26,27], since both found a significant association between low SMI with respectively acute dysphagia (CTCAE ≥ III) and late dysphagia (CTCAE ≥ II) in a large cohort of HNSCC patients treated with definitive (C)RT [26,27]. Several underlying mechanisms could explain the relationship between low SMI and toxicity. In the case of chemotherapy, low lean body mass (which largely corresponds to low SMI) results in less body volume, consequently risking chemotherapy overdose [47]. Therefore, to prevent chemotherapy-related toxicities, others advocate to adjust chemotherapy doses based on lean body mass instead of body surface area or weight [45,47]. Increased radiation-induced toxicities in patients with low SMI could also be due to a pro-inflammatory state [48,49] or due to malnutrition as a result of dysphagia [50,51]. Furthermore, low SMI is associated with impaired muscle function [15] and may also influence the swallowing muscles [51]. Accordingly, pre-treatment low SMI seems to have sufficient discriminative power to identify patients at risk for the occurrence of all-cause toxicities and dysphagia in HNSCC patients.

The present study could not significantly link low SMI and all-cause postoperative complications and wound infection (both defined as CDC grade ≥ II). Bril et al. included patients undergoing a total laryngectomy and found that low SMI was related to pharyngocutaneous fistulas and overall survival but not to all-grade (CDC ≥ I) or severe postoperative complications (CDC ≥ III) [52], which is in line with this present study. However, Bril et al. applied the non-sex-specific SMI cut-off values based on the occurrence of chemotherapy dose-limiting toxicity from Wendrich et al., to identify patients with low SMI in a surgical population, and may have resulted in the lack of a significant relationship between low SMI and postoperative complications, as graded with the CDC. Although two other studies, also using CDC (grade ≥ III), did find a relationship between low SMI and severe postoperative complication, though it was in patients undergoing mandibular reconstruction with a free fibula flap [53] and autologous head and neck free tissue reconstruction [54]. Outcomes of three the above-mentioned studies were further analysed in a meta-analysis and showed that low SMI was indeed related to postoperative complications [8]. Discrepancy between this meta-analysis and our study could be explained by differences in applied SMI cut-off values and/or endpoints for postoperative complications. We also performed additional analysis on infectious postoperative complications, as it has been found to be related to low SMI in other studies, so muscle loss is thought to be the result of reduced immune function, due to a decrease in skeletal muscle mass and increase in adipocytes [23]. However, this present study failed to find a significant relationship. Moreover, the relation between SMI and postoperative complications after less-invasive surgical interventions (e.g., transoral robotic surgery) remains unclear in HNSCC [55].

This present cohort had some obvious limitations. The oncological stage was determined according to 7th instead of 8th edition of American Joint Committee on Cancer Manual, as during the prospective inclusion of patients the newest 8th edition did not exist yet. It was not possible to stage according to the 8th edition, due to lacking data including invasion depth of oral cavity tumours, which could be seen as a limitation. Moreover, this study is very likely underpowered due to the limited sample size, especially in the surgical group. For some variables in the logistic regression with postoperative complications as the endpoint, ORs were unrealistically high or could not be calculated, as a result of too few patients within the category. Translation of the results regarding the surgical group is, therefore, difficult and is potentially biased due to heterogeneity in the tumour site and surgical procedure. Interpreting these data may have led to low AUCs in the surgical group and, therefore, underestimation of the effect and insignificant results. Unfortunately, performing subgroup analysis on specific types of surgery was not possible in this study, due to the low number of cases. Although higher AUCs were achieved in the (C)RT group, AUCs were still <0.7, indicating insufficient diagnostic discriminative power of SMI on toxicities. Low AUCs could also be due to heterogeneity within the (C)RT group as multiple treatment types were included, i.e., accelerated and not accelerated RT, CRT and RT with cetuximab. Moreover, a limited sample of variables could be included in the multivariable regression analysis due to the risk of overfitting. Ideally, more determinants such as frailty, comorbidities and treatment specifications (e.g., chemotherapy type and dosage) were included in the multivariable regression to analyse potential confounding. Nevertheless, low SMI, according to the generated SMI cut-off values, was found to be independently and significantly related to all-cause toxicity and dysphagia. Future research should be focused on the external validation of the here generated sex-specific SMI cut-off values.

Compared to prior publications regarding SMI cut-off values in HNSCC patients, the main strength is that this present study provided sex-specific cut-off values based on the occurrence of adverse events using ROC-curves, which is seen as the optimal method for defining clinically implacable cut-off values. Not only sex- and modality-specific SMI thresholds were generated in this present study but also the relationship of low SMI with adverse outcome was analysed, which is highly clinically relevant. Moreover, SMI measurements made for this analysis were reliable, as observer analysis showed excellent performance of the main observer. Furthermore, patients were prospectively included and assessed with a geriatric assessment at baseline, using a broad range of validated scales and tools. After inclusion, patients were systematically followed-up to assess the occurrence of adverse events, as measured with the CTCAE and CDC, resulting in a well-documented and reliable data-biobank.

## 5. Conclusions

To conclude, identifying HNSCC patients at risk for adverse events is highly clinically relevant. The sex-specific SMI cut-off values to diagnose low SMI, as generated in the (C)RT group, were able to identify patients at risk for all-cause toxicity and dysphagia. More research is necessary to establish SMI cut-off values based on postoperative complications, ideally in a larger and more homogenous HNSCC population. These outcomes help with constructing a practical and clinically relevant biomarker for adverse events in HNSCC patients.

## Figures and Tables

**Figure 1 jcm-11-04650-f001:**
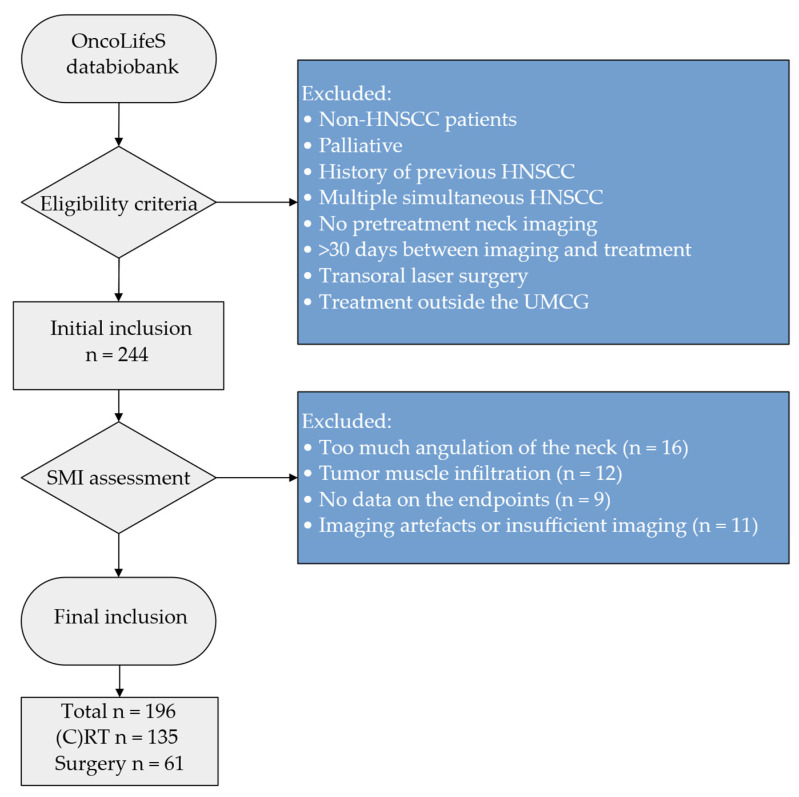
Flowchart of in- and excluded patients. Abbreviations: HNSCC = head and neck squamous cell carcinoma, (C)RT = (Chemo-)Radiotherapy, SMI = skeletal muscle index, UMCG = University Medical Center Groningen.

**Figure 2 jcm-11-04650-f002:**
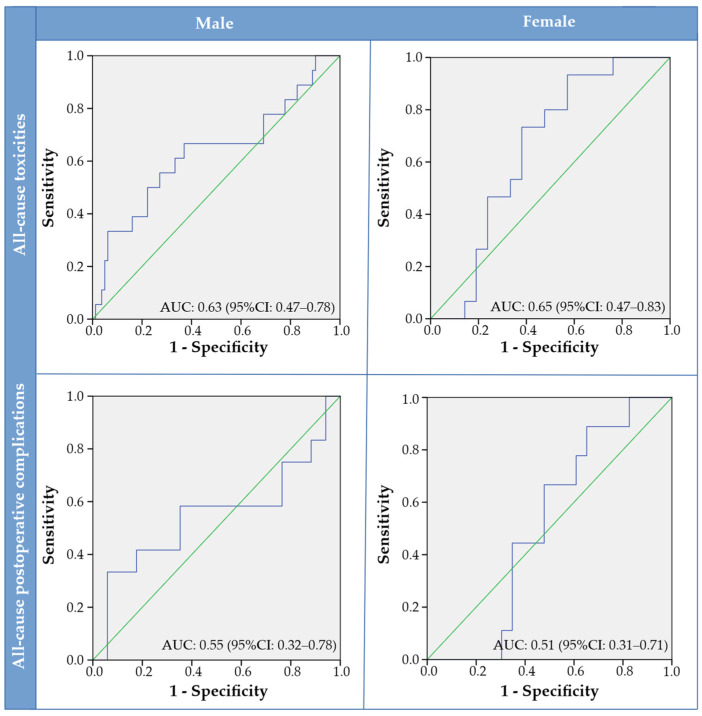
Receiver operator characteristic curves stratified for males versus females and all-cause toxicities versus all-cause postoperative complications. Abbreviations: AUC = area under the curve, CI = confidence interval.

**Table 1 jcm-11-04650-t001:** Baseline characteristics of total cohort and differences between patients treated with (chemo-)radiotherapy and surgery.

Characteristics	Total	(Chemo-) Radiotherapy	Surgery	*p*-Value
		n = 196	n = 135	n = 61
**Sex**				
** Male**	128 (65.3%)	99 (73.3%)	29 (47.5%)	**<0.001 ***
** Female**	68 (34.7%)	36 (26.7%)	32 (52.5%)	
**Age (years)**	64.2 (±11.2)	63.5 (±10.5)	65.8 (±12.5)	0.176 **
**BMI (kg^2^/m^2^)**	25.8 (±5.3)	25.4 (±5.4)	26.5 (±5.0)	0.184 **
**Tumour site**				
** Oropharynx**	69 (35.2%)	67 (49.6%)	2 (3.3%)	**<0.001 *****
** Oral cavity**	60 (30.6%)	5 (3.7%)	55 (90.2%)	
** Larynx**	56 (28.6%)	52 (38.5%)	4 (6.6%)	
** Hypopharynx**	10 (5.1%)	10 (7.4%)	0 (0%)	
** Nasopharynx**	1 (0.5%)	1 (0.7%)	0 (0%)	
**Stage**				
** I**	23 (11.7%)	3 (2.2%)	20 (32.8%)	**<0.001 ***
** II**	24 (12.2%)	17 (12.6%)	7 (11.5%)	
** III**	37 (18.0%)	31 (23.0%)	6 (9.8%)	
** IV**	112 (57.1%)	84(62.2%)	28 (45.9%)	
**HPV (Oropharynx)**				
** +**	22 (40.7%)	21 (39.6%)	1 (100%)	0.223 *
** −**	32 (59.3%)	32 (60.4%)	0 (0%)	
** Missing**	15	14	1	
**ACE-27**				
** None**	32 (28.3%)	24 (28.6%)	8 (37.6%)	0.113 *
** Mild**	37 (32.7%)	32 (38.1%)	5 (17.2%)	
** Moderate**	30 (26.5%)	20 (23.8%)	10 (34.5%)	
** Severe**	14 (12.4%)	8 (9.5%)	6 (20.7%)	
** Missing**	83	51	32	
**Smoking**				
** Never**	9 (8.8%)	3 (4.0%)	6 (22.2%)	**0.004 ***
** Active/quit**	93 (91.2%)	72 (96.0%)	21 (77.8%)	
** Missing**	94	60	34	
**Alcohol**				
** No abuse**	72 (77.4%)	51 (73.9%)	21 (87.5%)	0.170 *
** Abuse**	21 (22.6%)	18 (26.1%)	3 (12.5%)	
** Missing**	103	66	37	
**GFI**				
** Non-frail**	67 (69.1%)	50 (71.4%)	18 (64.3%)	0.419 *
** Frail**	30 (30.9%)	20 (28.6%)	10 (35.7%)	
** Missing**	99	65	34	
**G8**				
** Non-frail**	41 (42.7%)	32 (46.4%)	9 (33.3%)	0.245 *
** Frail**	55 (57.3%)	37 (53.6%)	18 (66.7%)	
** Missing**	100	66	34	

Significant *p*-values (<0.05) are indicated in bold. * Chi-squared test. ** Independent *t*-test. *** Chi-squared test was performed on a reorganised variable (oral cavity, nasopharynx/oropharynx and hypopharynx/larynx). Abbreviations: ACE-27 = Adult Comorbidity Evaluation-27, BMI = Body Mass Index, G8 = Geriatrics 8, GFI = Groningen Frailty Indicator, HPV = herpes papillomavirus.

**Table 2 jcm-11-04650-t002:** Occurrence of (chemo)radiotherapy toxicities and postoperative complications as measured with CTCAE grade ≥ III and Clavien–Dindo grade ≥II, respectively.

Characteristics	(Chemo-)Radiotherapy Toxicities	
	All-cause (n = 135)	Dysphagia (n = 135)
**CTCAE**		
** None**	8 (5.9%)	40 (29.6%)
** Grade I**	45 (33.3%)	36 (26.7%)
** Grade II**	49 (36.3%)	30 (22.2%)
** ≥Grade III**	33 (24.4%)	29 (21.5%)
**Characteristics**	**Postoperative complications**	
	All-cause (n = 61)	Wound infection (n = 61)
**Clavien–Dindo**		
** None**	32 (52.5%)	51 (78.5%)
** Grade I**	8 (13.1%)	3 (4.6%)
** Grade II**	12 (19.7%)	7 (10.8%)
** Grade III**	9 (14.8%)	4 (6.2%)
** ≥Grade II**	21 (34.4.0%)	11 (16.9%)

Abbreviations: CTCAE = Common Terminology Criteria for Adverse Events.

**Table 3 jcm-11-04650-t003:** SMI cut-off values based on the occurrence of adverse events endpoints. AUC, sensitivity and specificity per SMI cut-off value are provided.

	SMI Cut-Off	Sensitivity	Specificity	AUC	95%CI	*p*-Value
**Toxicities**						
**All-cause**						
**Male**	46.49	0.67	0.63	0.632	0.47–0.78	0.082
**Female**	37.90	0.93	0.43	0.648	0.47–0.83	0.136
**Dysphagia**						
**Male**	46.49	0.65	0.62	0.610	0.45–0.77	0.156
**Female**	34.91	0.75	0.58	0.642	0.46–0.83	0.169
**Postoperative complications**						
**All-cause**						
**Male**	41.97	0.33	0.94	0.549	0.32–0.78	0.658
**Female**	39.09	0.89	0.65	0.512	0.31–0.71	0.917
**Wound infection**						
**Male**	40.79	0.14	0.96	0.409	0.15–0.67	0.476
**Female**	35.64	1.00	0.54	0.607	0.42–0.79	0.494

Abbreviations: AUC = area under the curve, CI = confidence interval, SMI = skeletal muscle index (cm^2^/m^2^).

**Table 4 jcm-11-04650-t004:** Univariable logistic regression for individual predictors of all-cause toxicities and dysphagia toxicity according to CTCAE classification (grade ≥ III).

Characteristics	(Chemo-)Radiotherapy Toxicities					
	All-Cause			Dysphagia		
	OR	95% CI	*p*-Value	OR	95% CI	*p*-Value
**Sex**						
** Male**	Ref			Ref		
** Female**	0.31	0.14–0.72	**0.006**	0.42	0.17–0.99	**0.047**
**Age (years)**	0.97	0.93–1.00	0.076	0.97	0.93–1.01	0.126
**BMI (kg/m^2^)**	0.84	0.75–0.93	**0.001**	0.85	0.77–0.95	**0.003**
**Tumour site**						
** Oral cavity**	Ref		**0.007**	Ref		**0.011**
** Nasopharynx/Oropharynx**	0.83	0.13–5.24		1.91	0.20–18.14	
** Hypopharynx/Larynx**	0.19	0.03–1.35		0.43	0.04–4.48	
**Stage**						
** I–III**	Ref		**0.010**	Ref		**0.014**
** IV**	3.55	1.35–9.34		3.68	1.30–10.38	
**HPV (oropharynx)**						
** +**	Ref			Ref		
** −**	0.73	0.23–2.31	0.593	0.76	0.26–2.65	0.833
**ACE-27**						
** None**	Ref		0.704	Ref		0.837
** Mild**	1.24	0.20–7.67		1.00	0.16–6.35	
** Moderate**	1.17	0.20–6.94		1.00	0.14–5.99	
** Severe**	0.53	0.07–3.98		0.54	0.07–3.98	
**GFI**						
** Non-frail**	Ref		0.265	Ref		0.482
** Frail**	1.91	0.61–5.95		1.52	0.47–4.88	
**G8**						
** Non-frail**	Ref		0.621	Ref		0.810
** Frail**	1.32	0.44–4.01		1.15	0.37–3.54	
**Smoking**						
** Never**	Ref			Ref		
** Active/quit**	0.67	0.06–7.80	0.747	0.62	0.05–7.25	0.702
**Alcohol**						
** No abuse**	Ref			Ref		
** Abuse**	0.53	0.13–2.11	0.336	0.59	0.15–2.35	0.449
**SMI (cm^2^/m^2^)**	0.91	0.86–0.96	**<0.001**	0.92	0.87–0.97	**0.002**
**Low SMI**						
** No**	Ref			Ref		
** Yes**	3.86	1.69–8.86	**0.001**	2.70	1.12–6.47	**0.026**

Significant *p*-Values (<0.05) are indicated in bold. Abbreviations: ACE-27 = Adult Comorbidity Evaluation-27, BMI = Body Mass Index, CI = confidence interval, G8 = Geriatrics 8, GFI = Groningen Frailty Indicator, HPV = herpes papillomavirus, OR = odds ratio, SMI = skeletal muscle index.

**Table 5 jcm-11-04650-t005:** Univariable logistic regression for individual predictors of all-cause postoperative complications and postoperative wound infection according to Clavien–Dindo classification (grade ≥ II).

Characteristics	Postoperative Complications					
	All-Cause			Wound Infection		
	OR	95%CI	*p*-Value	OR	95%CI	*p*-Value
**Sex**						
** Male**	Ref			Ref		
** Female**	1.80	0.62–5.25	0.279	2.23	0.58–8.59	0.245
**Age (years)**	0.97	0.93–1.02	0.188	0.99	0.94–1.04	0.616
**BMI (kg/m^2^)**	0.96	0.86–1.07	0.479	1.01	0.89–1.15	0.830
**Tumour site**						
** Oral cavity**	Ref		0.827	Ref		0.026
** Nasopharynx/Oropharynx**	2.44	0.14–41.40		6.86	0.38–122.52	
** Hypopharynx/Larynx**	*	*		20.57	1.87–226.32	
**Stage**						
** I–III**	Ref		0.463	Ref		0.060
** IV**	1.49	0.52–4.30		4.00	0.95–16.93	
**HPV (oropharynx)**						
** +**	*			*		
** −**						
**ACE-27**						
** None**	Ref		0.303	Ref		0740
** Moderate**	0.67	0.07–6.87		1.67	0.12–24.26	
** Severe**	5.33	0.50–127.90		1.25	0.06–26.87	
** Missing**	1.33	0.24–16.36		3.33	0.28–40.29	
**GFI**						
** Non-frail**	Ref		0.517	Ref		0.974
** Frail**	1.69	0.35–8.22		0.97	0.18–537	
**G8**						
** Non-frail**	Ref		0.069	Ref		0.454
** Frail**	5.50	0.88–34.46		1.75	0.28–11.15	
**Smoking**						
** Never**	*	*		*	*	
** Active/quit**						
**Alcohol**						
** No abuse**	*	*		*	*	
** Abuse**						
**SMI (cm^2^/m^2^)**	1.02	0.96–1.08	0.529	1.04	0.96–1.11	0.344
**Low SMI (sex-specific)**						
** No**	Ref		0.377	Ref		
** Yes**	0.62	0.21–1.81		0.67	0.17–2.58	0.562
**Surgical intensity**						
** Minor**	Ref		0.287	Ref		0.148
** Major**	1.80	0.61–5.28		2.89	0.69–12.17	

Significant *p*-values (<0.05) are indicated in bold. * No regression was performed due to too few patients within the category. Abbreviations: ACE-27 = Adult Comorbidity Evaluation-27, BMI = Body Mass Index, CI = confidence interval, G8 = Geriatrics 8, GFI = Groningen Frailty Indicator, HPV = herpes papillomavirus, OR = odds ratio, SMI = skeletal muscle index.

**Table 6 jcm-11-04650-t006:** Multivariable regression analysis for determining predictors for all-cause (C)RT toxicities and dysphagia toxicity.

Characteristics	(Chemo-)Radiotherapy Toxicities					
	All-Cause			Dysphagia		
	OR	95%CI	*p*-Value	OR	95%CI	*p*-Value
**Low SMI**						
** No**	Ref		**0.006**	Ref		**0.046**
** Yes**	3.33	1.41–7.82		2.50	1.02–6.14	
**Stage**						
** I–III**	Ref		**0.014**	Ref		**0.020**
** IV**	3.45	1.28–9.29		3.47	1.21–9.92	

Age, tumour site, oncological stage and low SMI were included into the model. Tumour site and age were excluded when backward selection was applied. Significant *p*-values (<0.05) are indicated in bold. Abbreviations: BMI = Body Mass Index, CI = confidence interval, (C)RT = (chemo)radiotherapy, OR = odds ratio.

## Data Availability

Data are available on request due to privacy restrictions.

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
