# Peer review of "Sex-Specific Cut-Off Values for Low Skeletal Muscle Mass to Identify Patients at Risk for Treatment-Related Adverse Events in Head and Neck Cancer"

_jcm, 2022, doi:10.3390/jcm11164650_

Round 1
Reviewer 1 Report
In this manuscript, Zwart and colleagues retrospectively investigate 196 head and neck cancer patients using SMI at the C3 level to determine specific cut-off points to predict chemoradiotherapy toxicity and postsurgical complications. They analyzed these cutoff values in the context of both head and neck surgery and chemoradiotherapy. This is an important area of topic in the field of head and neck oncology for which the authors have published several articles previously. There are many important points that must be addressed.
Major Comments/Concerns:
1. Although the manuscript is consistent with the verbiage within this manuscript, I would suggest using the term “sex” and “sex-specific” rather than “gender”, as the former refers to biologic characteristics and latter refers to a self-defined expression. Furthermore, the term sex is binary, whereas gender has numerous categories and definitions, many of which overlap and difficult to distinguish. This many have not been much of an issue in years past but has become a point of contention recently to some readers.
2. Lines 54-55, there should be mention that head and neck cancer patients generally do not have abdominal imaging except in the setting of metastatic workup.
3. Line 70-72 – There are no citations for the claim of this sentence regarding improving skeletal muscle to improve outcomes. This is especially important in the context of cancer patients who have limited time before surgery and/or chemotherapy. I would recommend revising or removing this sentence.
4. Line 94 – further define “early-stage tumours”, as it is unclear if this means just in situ.
5. Lines 109-111 – Why is the 7th edition of the AJCC being used for staging when the current staging is the 8th edition? This is particularly important regarding nodal staging and HPV-oropharyngeal squamous cell carcinoma. These stages need to be updated, and if not, that needs to be explained explicitly and listed as a limitation.
6. In Table 1, for the category “Tumour Site”, there are 5 categories listed, five of which have less than 5 incidences and two of which have zero incidence. You cannot perform a Chi square test with that and therefore categories must be re-organized or removed and will alter your p-value. Additionally, there is listed two asterisks (**) after the p<0.001 indicating this is an independent t-test, which is not true and should instead be one asterisk.
7. My greatest criticism with this investigation is the inclusion of surgery toxicities. First, surgery and CRT are incredibly different and have drastic differences in risks and consequences, and really the two cannot be combined in terms of assessing toxicities. Second, only n=61 patients were included for surgery, which is incredibly underpowered although acknowledged in the discussion, and the authors did not perform a power analysis. Furthermore, the authors perform no description of these surgeries nor include the many other confounding factors that affect postoperative outcomes. For instance, there is a drastic difference in postoperative risk between a simple neck dissection versus a total laryngopharyngectomy with free flap reconstruction. Therefore, I would suggest that the inclusion of surgical patients and surgical complications be completely removed unless able to include many more patients and account for the numerous confounding variables of surgery where are not listed in this current version. There are also several articles already published (and referenced in this manuscript) analyzing low SMI in head and neck cancer surgery regarding postoperative complications, and this does not improve the literature on the subject.
8. It is unclear how the authors selected which variables be included in the multivariate regression for CRT toxicities. The methods state that variables were selected from the univariate regression and then placed in the multivariate regression, after which a backward step-wise method was used to determine the “final” model. The author used tumor site, TNM stage, and low SMI in the initial multivariate model and resulted with just SMI and TNM staging. Many other factors contribute to toxicity that the authors did not include and have been demonstrated in prior reports, particularly comorbidities (ACE-27), age, and frailty. Some that would be particularly important in this context yet were not included were the total dose of radiation and chemotherapy and agents used, as that would almost assuredly be the strongest predictor of toxicities, and once accounted may eliminate SMI or the TNM staging identified in multivariate regression. Although there were some variables insignificant on univariate analysis, the authors must explain how they are not included here simply because of a p-value. There are much more sound and sophisticated ways to balance the covariates to determine what is the appropriate cutoff for these adverse events, including propensity score matching.
9. Another issue with the statistical methods is the multivariate regression uses all-cause toxicities as the dependent variable. However, based on the listings in table 2, all patients except 8 (5.9%) of the CRT groups experienced at least a grade I toxicity. The binary logistic regression is therefore difficult to make sense if almost everyone encountered a toxicity. It would make more sense to have the endpoint be grade III or greater toxicities, as this is cited as clinically relevant in line 141. If this is not the case and the authors only used Grade III or greater as their endpoint, then it is not clear in the methods or results and should be clarified further.
10. Despite the conclusion that low SMI is such an important predictor of postoperative outcomes based on the limited, selected variables in their regression, the AUC generated were all quite poor and <0.70 as the authors cited is used as “clinically significant”. I believe this is from not including important variables in the regression as mentioned previously and suggests that using C3 SMI may not be as clinically useful in HNSCC. The discussion also states that using the AUC ROC method is optimal for determining a cutoff, yet their ROC is not impressive nor reaching the “significant” mark. The male and female curves both cross the 0.50 line at some point within the curve, and the sensitivity/specificities of the cut-offs listed are also low and quite variable in range for the different toxicities. This must be addressed in the discussion clearly, as it is only briefly mentioned in the first paragraph of the discussion. A larger cohort and better data collection may improve this and/or alter the conclustions.
11. There are several paragraphs dedicated to the discussion of SMI and cutoff values. It is interesting that each cancer and treatment have developed their own SMI cutoff values for different procedures and chemo(radio)therapies. There are some articles which have written about what a “normal” SMI is based of a healthy population and divided into sex and age brackets. It may be interesting to use those values or compare those to your own.1,2
Minor Comments/Concerns:
1. Line 54 – the word “Since” can be removed.
2. Lines 67-69 – There does not need to be a comma after “SMI” on line 69.
3. Lines 102-103 – is there a citation for the international guidelines, e.g., the NCCN guidelines?
4. Line 151 – “continues data” should be “continuous data”.
5. Lines 186-188 – Eliminate “explains that most patients were treated with (C)RT”, as many/most stage III/IV disease undergo both surgery and adjuvant CRT or surgery alone.
6. In Table 2, the lines with grade IV and grade V toxicities should be removed since there are zero incidence of these. Instead, the Grade III, IV, and V should be combined to say “≥Grade III” to remove wasted space.
7. Line 213 – Change the cm2/m2 to cm2/m2 to include a superscript.
References:
1. van der Werf A, Langius JAE, de van der Schueren MAE, et al. Percentiles for skeletal muscle index, area and radiation attenuation based on computed tomography imaging in a healthy Caucasian population. Eur J Clin Nutr. 2018;72(2):288-296. doi: 10.1038/s41430-017-0034-5.
2. Derstine BA, Holcombe SA, Ross BE, Wang NC, Su GL, Wang WC.Sci Rep. 2018;8(1):11369. doi: 0.1038/s41598-018-29825-5
Author Response
Dear reviewers and editors of the Journal of Clinical Medicine,
We thank you for the opportunity of submitting a revised version of our manuscript, entitled “Generate Sex-Specific Cut-Off Values for Low Skeletal Muscle Mass To Identify Patients At Risk For Treatment-Related Adverse Events In Head And Neck Cancer”, for publication as an original contribution to Journal of Clinical Medicine.
We have thoroughly read the reviewers’ comments and a point-by-point response to all comments will follow in this letter. Please find a marked copy of the revised manuscript.
We hope we fully responded to the comments of the reviewers and that you will find this manuscript suitable for publication in Journal of Clinical Medicine. All of the authors are aware of and agree to the content of the paper and their being listed as an author on the paper. We are looking forward to your comments again.
On behalf of the authors,
Sincerely,
Aniek Zwart, MD
Point-by point response:
We would like to thank all the reviewers for their time investment, critical reading and insightful comments on the manuscript. We followed the suggestions of the reviewers, and we believe our paper has improved substantially.
Reviewer #1
Major issue 1
Although the manuscript is consistent with the verbiage within this manuscript, I would suggest using the term “sex” and “sex-specific” rather than “gender”, as the former refers to biologic characteristics and latter refers to a self-defined expression. Furthermore, the term sex is binary, whereas gender has numerous categories and definitions, many of which overlap and difficult to distinguish. This many have not been much of an issue in years past but has become a point of contention recently to some readers.
We thank the reviewer for addressing this matter, we have changed “gender” into “sex”.
Major issue 2
Lines 54-55, there should be mention that head and neck cancer patients generally do not have abdominal imaging except in the setting of metastatic workup.
We have adjusted this sentence as requested.
Major issue 3
Line 70-72 – There are no citations for the claim of this sentence regarding improving skeletal muscle to improve outcomes. This is especially important in the context of cancer patients who have limited time before surgery and/or chemotherapy. I would recommend revising or removing this sentence.
We provided additional information with references in the manuscript regarding this topic.
“The latter is underscored by a previous interventional study, as they were able to maintain skeletal muscle mass with preoperative exercise in patients with pancreatic cancer [PMID: 33870744),]. However, the role of interventional studies on maintaining or improving skeletal muscle mass in HNSCC remains unclear, although pre-treatment interventions were found to be related to an improved quality of life [29] and were found to be feasible [PMID: 30400971].” Moreover, we agree with the reviewer that pre-treatment time is limited, and should ideally not be delayed as this is related to an increased recurrence risk (PMID: 35680075). Alternatively, an possible solution is to not only focus on pre-treatment interventions but also peri-treatment interventions.
Major issue 4
Line 94 – further define “early-stage tumours”, as it is unclear if this means just in situ.
We have specified early-stage tumours treated with intraoral laser surgery in this sentence.
“Exclusion criteria were patients with a history of previous HNSCC, multiple simultaneous (head and neck and non-head and neck) malignant tumours, early stage tumours (including stage I-II HNSCC and carcinoma in situ) treated by transoral laser surgery (because of its minimally invasive nature), treatment outside the UMCG, and an interval of more than 30 days between pre-treatment imaging and treatment.”
Major issue 5
Lines 109-111 – Why is the 7th edition of the AJCC being used for staging when the current staging is the 8th edition? This is particularly important regarding nodal staging and HPV-oropharyngeal squamous cell carcinoma. These stages need to be updated, and if not, that needs to be explained explicitly and listed as a limitation.
We have mentioned this as a limitation in the discussion.
“Oncological stage was determined according to 7th instead of 8th edition of American Joint Committee on Cancer Manual, as during the prospective inclusion of patients the newest 8th edition did not exists yet. It was not possible to stage according to the 8th edition due to lacking data including invasion depth of oral cavity tumours, and could be seen as a limitation.”
Major issue 6
In Table 1, for the category “Tumour Site”, there are 5 categories listed, five of which have less than 5 incidences and two of which have zero incidence. You cannot perform a Chi square test with that and therefore categories must be re-organized or removed and will alter your p-value. Additionally, there is listed two asterisks (**) after the p<0.001 indicating this is an independent t-test, which is not true and should instead be one asterisk.
The reviewer is correct that analysis cannot be performed with cells having zero incidence. The Chi Square test was performed on the re-organised variable of tumour site, but we had not specified that. Therefore, we have specified that in the legenda of table 1.
Major issue 7
My greatest criticism with this investigation is the inclusion of surgery toxicities. First, surgery and CRT are incredibly different and have drastic differences in risks and consequences, and really the two cannot be combined in terms of assessing toxicities. Second, only n=61 patients were included for surgery, which is incredibly underpowered although acknowledged in the discussion, and the authors did not perform a power analysis. Furthermore, the authors perform no description of these surgeries nor include the many other confounding factors that affect postoperative outcomes. For instance, there is a drastic difference in postoperative risk between a simple neck dissection versus a total laryngopharyngectomy with free flap reconstruction. Therefore, I would suggest that the inclusion of surgical Of the surgical group, 29 patients had transoral procedure with or without sentinel node excision, 15 patients had a commando procedure, 10 patients had a transoral procedure with any kind of neck dissection or flap reconstruction, 4 patients had a total laryngectomy, and 3 patients had a hemi- or partial maxillectomy. patients and surgical complications be completely removed unless able to include many more patients and account for the numerous confounding variables of surgery where are not listed in this current version. There are also several articles already published (and referenced in this manuscript) analyzing low SMI in head and neck cancer surgery regarding postoperative complications, and this does not improve the literature on the subject.
We agree with the reviewer that the head and neck population treated with surgery is very different in relation to patients treated with radiotherapy. Therefore, in this manuscript, different SMI cut-off values were generated and data was stratified for treatment modality. Multiple previous published articles in high impact journals, also stratified their cohort of head and neck cancer according to treatment modality (REF: PMID: 34111770, PMID: 34673914, PMID: 33045628), which underscores the robustness of the applied method in the present study. The reviewer correctly mentioned that the postoperative risk differs between types of surgery, and should be further addressed. Therefore, we specified the surgical procedures, and stratified those into minor and major surgery.
“Surgical procedures were stratified into minor and major surgery. Transoral surgery with or without sentinel node procedure was considered minor. Major surgery was in case of commando procedure, total laryngectomy (TLE), hemi- or partial maxillectomy, and transoral surgery with any type of neck dissection or flap reconstruction.”
“Of the surgical group, 29 patients had minor surgery (transoral procedure with or without sentinel node excision n = 29) and 32 patients had major surgery (commando procedure n = 15, transoral procedure with any kind of neck dissection or flap reconstruction n = 10, total laryngectomy n = 4, and hemi- or partial maxillectomy n = 3).”
The newly created variable with surgical intensity was added to the logistic regression analysis and was found to be insignificantly related to postoperative complications. In this present study, no significant relationship between low skeletal muscle mass and postoperative complications could be found, and is probably due to the relatively small and heterogenic surgical cohort, as mentioned in the discussion. Nevertheless, we strongly believe that our manuscript aids the current literature, as we are the first to generate sex-specific SMI cut-off values in patients surgically treated for head and neck cancer. These sex-specific SMI cut-off values can be used in literature to perform external validation of the here generated SMI cut-off values.
Major issue 8
It is unclear how the authors selected which variables be included in the multivariate regression for CRT toxicities. The methods state that variables were selected from the univariate regression and then placed in the multivariate regression, after which a backward step-wise method was used to determine the “final” model. The author used tumor site, TNM stage, and low SMI in the initial multivariate model and resulted with just SMI and TNM staging. Many other factors contribute to toxicity that the authors did not include and have been demonstrated in prior reports, particularly comorbidities (ACE-27), age, and frailty. Some that would be particularly important in this context yet were not included were the total dose of radiation and chemotherapy and agents used, as that would almost assuredly be the strongest predictor of toxicities, and once accounted may eliminate SMI or the TNM staging identified in multivariate regression. Although there were some variables insignificant on univariate analysis, the authors must explain how they are not included here simply because of a p-value. There are much more sound and sophisticated ways to balance the covariates to determine what is the appropriate cutoff for these adverse events, including propensity score matching.
We have specified our methods selecting variables from the univariate regression to the multivariable regression analysis in the manuscript.
“Clinical relevant and significant variables from the univariate logistic regression with the highest impact on the AE endpoints according to their OR were selected as covariates for the multiple logistic regression, and covariates were excluded from the multiple model in a backwards manner.”
The reviewer mentioned that some possible other important clinical covariates for toxicity are missing including frailty and comorbidities. We agree that it’s important to analyse as many known related variables to understand associations and potential confounding. Frailty, as measured with the geriatrics 8 and Groningen Frailty Indicator, and comorbidities, according to the Adult Comorbidity Evaluation-27, were not significantly related acute toxicities in a previous published study (PMID: 34111770). Therefore, it seems that frailty and comorbidities are less important in relation to the occurrence of acute toxicities in head and neck cancer patients, and presumably less of importance in the multivariable regression analysis as performed in the present study. Moreover, we have relatively high rate of missing’s for Geriatrics 8, Groningen Frailty Indicator and Adult Comorbidity Evaluation-27 and therefore not included in the multivariate regression analysis due to potential bias. We added age as a potential important clinical variable to the multivariable model.
“For the multivariable regression analysis, tumour site, TNM stage, and low SMI were selected from the univariate logistic regression analysis according to their OR and p-values, and age was added to the model as a clinical relevant variable. Other important clinical relevant variables such as G8, GFI and ACE-27 were not included to the multivariable model due to a relatively high rate of missing’s.”
Major issue 9
Another issue with the statistical methods is the multivariate regression uses all-cause toxicities as the dependent variable. However , based on the listings in table 2, all patients except 8 (5.9%) of the CRT groups experienced at least a grade I toxicity. The binary logistic regression is therefore difficult to make sense if almost everyone encountered a toxicity. It would make more sense to have the endpoint be grade III or greater toxicities, as this is cited as clinically relevant in line 141. If this is not the case and the authors only used Grade III or greater as their endpoint, then it is not clear in the methods or results and should be clarified further.
In this manuscript, all-cause toxicities are corresponding to grade ≥III toxicities irrespective of toxicity domain, and not to grade ≥I toxicities. Because this was not clear according to our materials and methods, we have further specified this matter.
“Domains scored by the CTCAE were: xerostomia, taste, throat pain, oral pain, general pain, dysphagia, hoarseness and mucositis. All-cause (C)RT toxicity was defined as any CTCAE toxicity grade ≥ III, irrespective of domain. Dysphagia toxicity with CTCAE grade ≥ III was scored separately. “
Major issue 10
Despite the conclusion that low SMI is such an important predictor of postoperative outcomes based on the limited, selected variables in their regression, the AUC generated were all quite poor and <0.70 as the authors cited is used as “clinically significant”. I believe this is from not including important variables in the regression as mentioned previously and suggests that using C3 SMI may not be as clinically useful in HNSCC. The discussion also states that using the AUC ROC method is optimal for determining a cutoff, yet their ROC is not impressive nor reaching the “significant” mark. The male and female curves both cross the 0.50 line at some point within the curve, and the sensitivity/specificities of the cut-offs listed are also low and quite variable in range for the different toxicities. This must be addressed in the discussion clearly, as it is only briefly mentioned in the first paragraph of the discussion. A larger cohort and better data collection may improve this and/or alter the conclustions.
Thank you for addressing this major issue. Please see also major issue 8, in which we thoroughly address the selection of variables for the multivariable regression analysis. As mentioned in the present manuscript, the ROC-curves is the most optimal method to define SMI cut-off values, but AUC in this present study were not sufficient (<0.7), and the generated SMI cut-off values showed low specificity and sensitivity. It can be argued that the low AUCs are due to our variable selection (e.g. it may improve in a more homogenic surgical population), but is probably due to the low sample size in general. As requested by the reviewer we addressed this matter more clearly in the discussion. Currently, it is clearly stated in the key results of the discussion, and is further explained in the limitations.
“AUCs’ were all <0.7, indicating insufficient discriminative power of low SMI on the endpoints.”
“Interpreting this data may have led to low AUCs in the surgical group and therefore underestimation of the effect and insignificant results. Unfortunately, performing subgroup analysis on specific types of surgery was not possible in this study due to low number of cases. Although higher AUCs were achieved in the (C)RT group, AUCs were still <0.7 indicating insufficient diagnostic discriminative power of SMI on toxicities. Low AUCs could also be due to heterogeneity within the (C)RT group as multiple treatment types were included, i.e. accelerated and not accelerated RT, CRT and RT with targeted therapy.”
Major issue 11
There are several paragraphs dedicated to the discussion of SMI and cutoff values. It is interesting that each cancer and treatment have developed their own SMI cutoff values for different procedures and chemo(radio)therapies. There are some articles which have written about what a “normal” SMI is based of a healthy population and divided into sex and age brackets. It may be interesting to use those values or compare those to your own.1,2
HNC patients are, even in regards to other oncological populations, a very specific oncological population as they are eminently at risk for malnutrition due to the location of their tumours in the upper aero-gastrointestinal tract. Up to 60% of head and neck cancer patients are already malnourished before initiating therapy (PMID: 20737491, PMID: 20824806, PMID: 23845698). Therefore, to apply SMI cut-off values of a healthy population in a HNC population seems not scientifically reliable.
Minor issue 1
Line 54 – the word “Since” can be removed.
The word “since” has been removed from this sentence.
Minor issue 2
Lines 67-69 – There does not need to be a comma after “SMI” on line 69.
The comma has been removed from this sentence.
Minor issue 3
Lines 102-103 – is there a citation for the international guidelines, e.g., the NCCN guidelines?
We have referred to the NCCN guidelines.
Minor issue 4
Line 151 – “continues data” should be “continuous data”.
“Continues” has been changed in “continuous”.
Minor issue 5
Lines 186-188 – Eliminate “explains that most patients were treated with (C)RT”, as many/most stage III/IV disease undergo both surgery and adjuvant CRT or surgery alone.
This sentence has been removed as requested by the reviewer.
Minor issue 6
In Table 2, the lines with grade IV and grade V toxicities should be removed since there are zero incidence of these. Instead, the Grade III, IV, and V should be combined to say “≥Grade III” to remove wasted space.
Table 2 has been changed as requested by the reviewer.
Minor issue 7
Line 213 – Change the cm2/m2 to cm2/m2 to include a superscript.
This sentence has been changed as requested by the reviewer.
Reviewer 2 Report
to improve the discussion and quality:
- line 40, Head and neck carcinomas are often diagnosed at an advanced stage and for this reason the most proposed treatment remains chemoradiotherapy. however, the new less invasive methods allow patients to be treated with fewer side effects., please discuss and cite doi:10.1016/j.anl.2021.05.007
- A typical assessment for sarcopenia involves the use of abdominal computed tomography (CT) for calculating the skeletal muscle index (SMI) at the level of the third lumbar vertebra (L3). However, abdominal CT is not regularly performed on patients with head and neck cancer (HNC). Masticatory SMI (M-SMI) measurements based on head and neck CT scans can be used to conduct sarcopenia assessments by evaluating whether M-SMI is correlated with L3-SMI. please discuss and cite doi:10.1371/journal.pone.0251455
- strobe guidelines should be adopted
- consort diagram should be added
Author Response
Dear reviewers and editors of the Journal of Clinical Medicine,
We thank you for the opportunity of submitting a revised version of our manuscript, entitled “Generate Sex-Specific Cut-Off Values for Low Skeletal Muscle Mass To Identify Patients At Risk For Treatment-Related Adverse Events In Head And Neck Cancer”, for publication as an original contribution to Journal of Clinical Medicine.
We have thoroughly read the reviewers’ comments and a point-by-point response to all comments will follow in this letter. Please find a marked copy of the revised manuscript.
We hope we fully responded to the comments of the reviewers and that you will find this manuscript suitable for publication in Journal of Clinical Medicine. All of the authors are aware of and agree to the content of the paper and their being listed as an author on the paper. We are looking forward to your comments again.
On behalf of the authors,
Sincerely,
Aniek Zwart, MD
Point-by point response:
We would like to thank all the reviewers for their time investment, critical reading and insightful comments on the manuscript. We followed the suggestions of the reviewers, and we believe our paper has improved substantially.
Reviewer #2
Major issue 1
line 40, Head and neck carcinomas are often diagnosed at an advanced stage and for this reason the most proposed treatment remains chemoradiotherapy. however, the new less invasive methods allow patients to be treated with fewer side effects., please discuss and cite doi:10.1016/j.anl.2021.05.007 .
We have added a sentence regarding this issue in the discussion as requested by the reviewer.
“Moreover, the relation between SMI and postoperative complications after less invasive surgical interventions (e.g. transoral robotic surgery, doi:10.1016/j.anl.2021.05.007) remains unclear in HNSCC.”
Major issue 2
A typical assessment for sarcopenia involves the use of abdominal computed tomography (CT) for calculating the skeletal muscle index (SMI) at the level of the third lumbar vertebra (L3). However, abdominal CT is not regularly performed on patients with head and neck cancer (HNC). Masticatory SMI (M-SMI) measurements based on head and neck CT scans can be used to conduct sarcopenia assessments by evaluating whether M-SMI is correlated with L3-SMI. please discuss and cite doi:10.1371/journal.pone.0251455.
As requested by the reviewer, we added this topic and cited accordingly in the discussion.
“ However, other possible alternatives exists to measure SMI in HNSCC patients, for instance measurements on the masticatory muscles (REF: doi:10.1371/journal.pone.0251455.)”
Major issue 3
Strobe guidelines should be adopted
The Strobe Guidelines checklist was used and the manuscript was corrected if necessary.
Major issue 4
Consort diagram should be added
A flow chart of in- and excluded patients has been added to the manuscript.
Reviewer 3 Report
Thank you for the opportunity to review your manuscript about the Gender Specific Cut-Off Values for Low Skeletal Muscle Mass To Identify Patients At Risk For Treatment-Related Adverse Events In Head and Neck Cancer. Overall, I believe this study has potential scientific utility. However, there are some issues described as below. In addition, I think authors had better focus on relationship between CRT and SMI.
Major Comments:
1) What kinds of drug do Authors use for chemoradiotherapy?
Do Authors use high dose cisplatin? Authors should describe it in details.
In addition, are cisplatin dose association with CRT toxicities or
SMI?
2) Patients with early stage accounts for about 15% in CRT
groups.
Why did authors use CRT for patients with early stage, particular stage I.
I think including patients with early stage affects the number of adverse events.
3) Authors should describe adverse events in details. Authors described only
All-cause and Dysphagia, All-cause and Wound infection.
Please describe xerostomia, taste, throat pain, and radiation
mucositis, dermatitis. Especially, radiation mucositis and
dermatitis are very important.
4) The sample size in the surgical group is too small.
5) What kinds of surgery did the patient have? with
reconstruction? Please describe the surgical technique in
details.
Minor Comments:
In some tables, authors describes TNM.
Did this mean clinical stage?
Author Response
Dear reviewers and editors of the Journal of Clinical Medicine,
We thank you for the opportunity of submitting a revised version of our manuscript, entitled “Generate Sex-Specific Cut-Off Values for Low Skeletal Muscle Mass To Identify Patients At Risk For Treatment-Related Adverse Events In Head And Neck Cancer”, for publication as an original contribution to Journal of Clinical Medicine.
We have thoroughly read the reviewers’ comments and a point-by-point response to all comments will follow in this letter. Please find a marked copy of the revised manuscript.
We hope we fully responded to the comments of the reviewers and that you will find this manuscript suitable for publication in Journal of Clinical Medicine. All of the authors are aware of and agree to the content of the paper and their being listed as an author on the paper. We are looking forward to your comments again.
On behalf of the authors,
Sincerely,
Aniek Zwart, MD
Point-by point response:
We would like to thank all the reviewers for their time investment, critical reading and insightful comments on the manuscript. We followed the suggestions of the reviewers, and we believe our paper has improved substantially.
Reviewer #3
Major issue 1
What kinds of drug do Authors use for chemoradiotherapy? Do Authors use high dose cisplatin? Authors should describe it in details. In addition, are cisplatin dose association with CRT toxicities or SMI?
Thank you for addressing this lacking information, we specified the used (C)RT protocols in the materials and methods.
“In general, type of (C)RT treatment depended on oncological stage and patients age or fitness. Treatment options according to this protocol were: accelerated radiation therapy (6 fractions per week); conventional fractionated radiation therapy; concomitant chemoradiation (3 courses of carboplatin on day 1, 300-350 mg/m2 and 5-fluorouracil on day 1-4, 600 mg/m2/24 h every 3 weeks); 3 courses of cisplatin (100 mg/m2) on day 1, 22, and 43; accelerated radiation therapy with weekly cetuximab (400 mg/m2 1 week before radiation therapy, followed by weekly 250 mg/m2 during radiation therapy) (PMID: 34048816:).”
We agree with the reviewer that is very interesting to analyse the relation of chemotherapy specifications in relation to our endpoints. Unfortunately, our current database doesn’t have information about the chemotherapy type and dosage, and could not be added to the analysis. Therefore we have added this as a limitation in the discussion.
“Moreover, a limited sample of determinants could be included in the multivariable regression analysis. Ideally, more determinants such as frailty, comorbidities, and treatment specifications (e.g. chemotherapy type and dosage) were included in the multivariable regression to analyze potential confounding.”
Major issue 2
Patients with early stage accounts for about 15% in CRT groups. Why did authors use CRT for patients with early stage, particular stage I. think including patients with early stage affects the number of adverse events.
Whitin the (C)RT group patients received different types of treatment and patients age or fitness, see also the reply on major issue 1. In general, patients with early stage disease received accelerated radiation therapy (6 fractions per week) and not CRT. We specified in the results different types of treatment were given, and we referred to the used protocol in the materials and methods.
“Of the (C)RT group, 54 patients had concomitant chemoradiation, 40 patients had conventional radiotherapy, 34 patients had accelerated radiotherapy, and 7 patients had accelerated radiotherapy with cetuximab.”
“In general, type of (C)RT treatment depended on oncological stage and patients age or fitness. Treatment options according to this protocol were: accelerated radiation therapy (6 fractions per week); conventional fractionated radiation therapy; concomitant chemoradiation (3 courses of carboplatin on day 1, 300-350 mg/m2 and 5-fluorouracil on day 1-4, 600 mg/m2/24 h every 3 weeks); 3 courses of cisplatin (100 mg/m2) on day 1, 22, and 43; accelerated radiation therapy with weekly cetuximab (400 mg/m2 1 week before radiation therapy, followed by weekly 250 mg/m2 during radiation therapy) (PMID: 34048816:).”
Major issue 3
Authors should describe adverse events in details. Authors described only. All-cause and Dysphagia, All-cause and Wound infection. Please describe xerostomia, taste, throat pain, and radiation mucositis, dermatitis. Especially, radiation mucositis and dermatitis are very important.
As requested by the reviewer we added more details of other types of toxicity.
“Thirty-tree (24.4%) patients experienced clinically relevant toxicity (CTCAE ≥ III) irrespective of toxicity domain. On all toxicity domains no grade IV or V (death) toxicity occurred. Twenty-nine (21.5%) patients suffered dysphagia grade III or more. In other toxicity domains, prevalence of CTCAE grade ≥ III were 3.0% xerostomia, 3.0% hoarseness, 2.3% mucositis, 2.3% general pain, 0.8% oral pain, 0.8% throat pain. No grade ≥ III toxicity on the domain of taste occurred.”
Major issue 4
The sample size in the surgical group is too small.
Ideally we had a larger and more homogenic population of patients receiving surgery. Nevertheless, we strongly believe that our manuscript aids the current literature, as we are the first to generate sex-specific SMI cut-off values in patients surgically treated for head and neck cancer. These sex-specific SMI cut-off values can be used in literature to perform external validation of the here generated SMI cut-off values.
Major issue 5
What kinds of surgery did the patient have? with reconstruction? Please describe the surgical technique in details.
As requested by the reviewer we described the types of surgery in the results.
“Of the surgical group, 29 patients had minor surgery (transoral procedure with or without sentinel node excision n = 29) and 32 patients had major surgery (commando procedure n = 15, transoral procedure with any kind of neck dissection or flap reconstruction n = 10, total laryngectomy with or without neck dissection n = 4, and hemi- or partial maxillectomy n = 3).”
Minor issue 1
In some tables, authors describes TNM. Did this mean clinical stage?
The reviewer is correct this should be oncological stage, we have changed this correctly.
Round 2
Reviewer 3 Report
Thank you very much. The manuscript has been revised well. I appreciate the effort of the work.
Major comments 1
The lack of information about the chemotherapy type and dosage is unacceptable.
Because authors discuss a promising predictor for adverse events of CRT.
Major comments 2
Originally surgery and chemoradiotherapy are separate topic.
Authors should discuss surgery and chemoradiotherapy separately, not in same manuscript.
Major comment 3
There are multiple surgical technique in surgery group.
It is difficult to discuss adverse event with equivalence
Author Response
Dear third reviewer and editors of the Journal of Clinical Medicine,
Again, we would like to thank you for the opportunity of submitting a revised version of our manuscript, entitled “Generate Sex-Specific Cut-Off Values for Low Skeletal Muscle Mass To Identify Patients At Risk For Treatment-Related Adverse Events In Head And Neck Cancer”, for publication as an original contribution to Journal of Clinical Medicine.
We hope we fully responded to the comments of the third reviewer and the revised manuscript is now suitable for publication in Journal of Clinical Medicine. All of the authors are aware of and agree to the content of the paper and their being listed as an author on the paper. We are looking forward to your comments again.
On behalf of the authors,
Sincerely,
Aniek Zwart, MD
Reviewer #3 second round
Major comments 1
The lack of information about the chemotherapy type and dosage is unacceptable. Because authors discuss a promising predictor for adverse events of CRT.
We agree that chemotherapy type and dosage in each individual patient and its relation to the outcome would be interesting. However, we disagree that this is mandatory for our study, as this issue has been widely investigated in previous studies and previous published articles are indicating that chemotherapy specifications are not an important (confounding) predictor for chemotherapy toxicity in head and neck cancer patients (PMID: 28688687, PMID: 35565223, and PMID: 31085391). Wendrich et al. looked at the relation of SMI and dose-limiting toxicity in 112 head and neck cancer patients receiving CRT. Their multivariable regression analysis showed that BMI and SMI were independently related to dose-limiting toxicity, but chemotherapy dosage was not (PMID: 28688687). Moreover, Morse et al. couldn’t detect a significant relation between chemotherapy type and chemotherapy toxicity (defined as delays greater than 1 week in therapy administration or failure to complete all planned cycles of chemotherapy) in a large cohort of 272 head and neck cancer patients, but low SMI was found to be independent related to toxicity (PMID: 35565223). Moreover, Ganju et al. could also not significantly link chemotherapy type to chemotherapy toxicity (defined as delays greater than 1 week in chemotherapy administration or failure to complete all planned cycles of chemotherapy) in 246 head and neck cancer patients (PMID: 31085391). Furthermore, the made point of the reviewer reflects on only 54 patients treated with chemoradiation of the in total 196 included patients. As mentioned in the first rebuttal, unfortunately, our current database doesn’t have information about the chemotherapy type and dosage; therefore, it could not be added to the analysis. We have added this as a limitation in the discussion.
“Moreover, a limited sample of determinants could be included in the multivariable regression analysis. Ideally, more determinants such as frailty, comorbidities, and treatment specifications (e.g. chemotherapy type and dosage) were included in the multivariable regression to analyze potential confounding.”
Major comments 2
Originally surgery and chemoradiotherapy are separate topic. Authors should discuss surgery and chemoradiotherapy separately, not in same manuscript.
We agree with the reviewer that the head and neck population treated with surgery is very different in relation to patients treated with radiotherapy. However, we disagree, that these results cannot be published in one manuscript. Multiple previous published articles in high impact journals, also stratified their cohort of head and neck cancer according to treatment modality (PMID: 34111770, PMID: 34673914, PMID: 33045628), which underscores the robustness of the applied method in the present study. Therefore, in the present manuscript, different SMI cut-off values were generated and data was stratified for treatment modality. In our opinion, the present study setup, including all treatment modalities as in the clinical practice, treatment decision has to be made often before it is known what kind of treatment modality would be applied. Prediction of adverse events (regarding surgical complications and (chemo)radiation toxicity) is ideally done before initiating treatment.
Major comment 3
There are multiple surgical technique in surgery group. It is difficult to discuss adverse event with equivalence.
We agree with the reviewer, that this is another limitation of the study. Postoperative risk differs between types of surgery and was further addressed in the revised manuscript. We have specified the surgical procedures and stratified those into minor and major surgery.
“Surgical procedures were stratified into minor and major surgery. Transoral surgery with or without sentinel node procedure was considered minor. Major surgery was in case of commando procedure, total laryngectomy (TLE), hemi- or partial maxillectomy, and transoral surgery with any type of neck dissection or flap reconstruction.”
“Of the surgical group, 29 patients had minor surgery (transoral procedure with or without sentinel node excision n = 29) and 32 patients had major surgery (commando procedure n = 15, transoral procedure with any kind of neck dissection or flap reconstruction n = 10, total laryngectomy n = 4, and hemi- or partial maxillectomy n = 3).”
The newly created variable with surgical intensity was added to the logistic regression analysis and was found to be insignificantly related to postoperative complications. In this present study, no significant relationship between low skeletal muscle mass and postoperative complications could be found, and is probably due to the relatively small and heterogenic surgical cohort, as mentioned as a limitation in the discussion.
“Moreover, this study is very likely underpowered due to the limited sample size, especially in the surgical group. For some variables in the logistic regression with postoperative complications as the endpoint, ORs were unrealistic high or could not be calculated as a result of too few patients within the category. Translation of the results regarding the surgical group is therefore difficult and is potentially biased due to heterogeneity in the tumour site and surgical procedure. Interpreting this data may have led to low AUCs in the surgical group and therefore underestimation of the effect and insignificant results. Unfortunately, performing subgroup analysis on specific types of surgery was not possible in this study due to low number of cases.”
Nevertheless, we strongly believe that our manuscript aids the current literature, as we are the first to generate sex-specific SMI cut-off values in patients surgically treated for head and neck cancer. These sex-specific SMI cut-off values can be used in literature to perform external validation of the here generated SMI cut-off values.
